# Denoising Diffusion Probabilistic Models for High-Fidelity fMRI Intrinsic Connectivity Network Data Generation

Meenu Ajith[1] and Vince D. Calhoun[2]

*Abstract*— The emergence of diffusion models such as Glide, Dalle-2, Imagen, and Stable Diffusion marks a significant breakthrough in generative AI-based image generation. This paper introduces an rs-fMRI image synthesis framework that leverages the nonlinear capabilities of denoising diffusion probabilistic models (DDPMs) to overcome the limitations of linear methods like independent component analysis (ICA) in neuroimaging analysis. Unlike ICA, which assumes linearity, DDPMs capture the intricate and complex patterns inherent in neuroimaging data. Our approach advances from 2D to 3D representations, providing a comprehensive visualization of intrinsic connectivity networks (ICNs). This framework also addresses the challenge of sparse training datasets commonly encountered in deep learning applications for neuroimaging. Trained on a large database, our model captures the intricate variability of different ICNs, generating realistic connectivity patterns. The proposed method is evaluated quantitatively to compare the synthesized ICNs against ground truth data. Results demonstrate that the proposed DDPM-based framework showcases competitive performance accuracy in reflecting the true complexity of neural connectivity patterns.

*Index Terms: rs-fMRI data, generative modeling, image synthesis, diffusion model.*

## I. INTRODUCTION

In recent years, deep learning (DL) based methods for medical image synthesis, such as generative adversarial networks (GANs) [1], have shown remarkable success across various tasks, including image denoising [2], super-resolution [3], data augmentation [4], cross-modality image synthesis [5], and image-to-image translation [6]. However, GANs encounter challenges such as mode collapse, training instability, and a lack of interpretability. To tackle this issue, diffusion-based generative models [7] were introduced in 2015. These models gained popularity recently with the emergence of denoising diffusion probabilistic models (DDPMs) [8] and latent diffusion models (LDMs) [9]. These alternatives to GANs provide higher quality and more diverse synthetic images by employing U-nets [10] to learn denoising for image generation. Furthermore, by avoiding adversarial training, diffusion models also enhance training stability and produce more realistic images.

In a recent study, DDPM was used to generate 3D brain structural MRI images, and in a quantitative comparison, it outperformed a 3D-$\alpha$-Wasserstein GAN [11]. Another study employed a conditional latent DDPM for medical image generation, comparing it with GAN-based models using images from ophthalmology, radiology, and histopathology. They showed that DDPMs excelled over GANs in both precision and diversity [12]. These studies have demonstrated that DDPMs offer a promising alternative to GANs in the field of neuroimaging. In this work, we use DDPMs for high-quality intrinsic connectivity networks (ICNs) generation. In our work, ICNs are specific independent components (ICs) obtained through independent component analysis (ICA) that have been identified as representing meaningful and reproducible patterns of functional connectivity within the brain. Here, we generate 2D and 3D synthetic ICNs using the proposed model by training on real ICNs obtained from a pipeline called NeuroMark [13], which leverages an a spatially constrained ICA model [14]. While ICA is a powerful method for extracting ICs from neuroimaging data, using DDPMs offers several distinct advantages. DDPMs can generate many synthetic ICNs, which is valuable for augmenting limited datasets. This is particularly useful in neuroimaging, where acquiring new data can be expensive and time-consuming. DDPMs can be used to generate ICNs with controlled variability, allowing researchers to create datasets with specific characteristics or test hypotheses under various conditions. Moreover, DDPMs can effectively denoise data, potentially improving the quality of the ICNs generated compared to direct ICA results, which may still contain residual noise and artifacts. Finally, DDPMs are nonlinear models, whereas ICA assumes linearity. This nonlinearity allows DDPMs to capture more complex patterns in neuroimaging data that linear methods like ICA might miss.

## II. METHODS

### A. Data

The neuroimaging training dataset for this analysis was acquired from the UK Biobank database. It comprised 30,000 participants who had undergone rs-fMRI scanning using 3 Tesla Siemens Skyra scanners with 32-channel head coils. Preprocessing steps for the rs-fMRI data included motion correction with MCFLIRT, grand-mean intensity normalization, high-pass temporal filtering, and geometric corrections using FSL's Topup tool. Additionally, EPI unwarping and gradient distortion correction were applied. Artifacts were eliminated using ICA and FMRIB's ICA-based X-noiseifier, and the data were standardized to an MNI EPI template, followed by Gaussian smoothing with a 6mm FWHM. Subsequently, the NeuroMark ICA process was applied to the 4D-preprocessed rs-fMRI data, utilizing the

*This work was supported by the GSU RISE program and NSF grants 2316421 and 2316421.

[1]Meenu Ajith and [2]Vince Calhoun are with the Tri-Institutional Center for Translational Research in Neuroimaging and Data Science (TReNDS) @ Georgia State, Georgia Tech, and Emory University, Atlanta, GA 30303, USA, USA (e-mail: majith@gsu.edu; vcalhoun@gsu.edu).

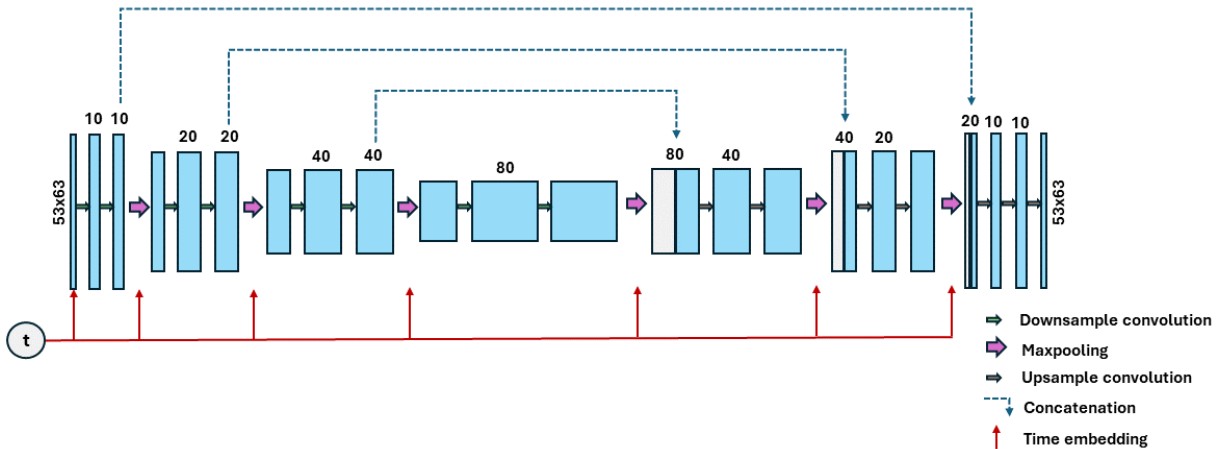

Fig. 1: Modified U-Net architecture of the DDPM model.This architecture has been tailored to enhance the model's ability to effectively add and remove noise from images across multiple scales, leveraging the U-Net's strengths in capturing fine details and broader contextual information simultaneously.

Neuromark_fMRI_1.0 template with 53 intrinsic connectivity networks (ICNs) derived from multiple blind ICA decompositions. These 53 ICNs were categorized into seven domains: subcortical (SC: 5 ICNs), auditory (AUD: 2 ICNs), sensorimotor (SM: 9 ICNs), visual (VIS: 9 ICNs), cognitive control (CC: 17 ICNs), default mode (DM: 7 ICNs), and cerebellar (CB: 4 ICNs). Six different ICNs were employed for training the proposed model.

### B. Denoising Diffusion Probabilistic Model

DDPMs form the foundation of current diffusion models in the field of generative AI. These models operate through two main steps: the forward noising process and the backward denoising process. In the forward noising process, DDPMs employ a Markov chain model with a predefined number of time steps, denoted as $T$. During each step, noise is incrementally introduced to the input image, starting at $t = 0$ and proceeding until $t = T$. The objective of this process is to transform the image into pure Gaussian noise by the final time step $T$. This transformation is governed by a noising function, represented by the conditional distribution $q(\mathbf{x}_t|\mathbf{x}_{t-1})$, which determines the noise addition at each step based on the image from the preceding time step. Noise schedulers are utilized to regulate the amount of noise added at each step.

The goal is to develop a generative model that can predict the noise added to an image at a specific timestamp. This is achieved through the concept of the backward process, utilizing the conditional distribution function $p(\mathbf{x}_{t-1}|\mathbf{x}_t)$. Directly modeling this function is impractical due to the vast number of possible images $\mathbf{x}_{t-1}$ for a given $\mathbf{x}_t$. Consequently, neural networks are employed for estimation, modifying the function to $p_{\boldsymbol{\theta}}(\mathbf{x}_{t-1}|\mathbf{x}_t, t)$ where $\boldsymbol{\theta}$ represents the network parameters. For this purpose, a U-Net architecture is employed, as illustrated in Fig.1.

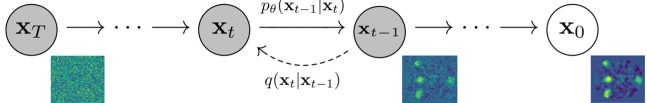

Fig. 2: Graphical model illustrating the DDPMs. This model depicts the process of gradually adding noise to an image and then reversing this process to denoise the image, effectively learning to generate high-quality images from random noise.

The modified U-Net architecture comprises three down-sampling parts, where each part typically includes a convolutional layer followed by a max-pooling operation, reducing the feature maps' spatial dimensions. The central bottleneck consists of multiple convolutional layers, reducing the number of feature maps. Following the bottleneck, there are three up-sampling steps, each consisting of a transposed convolutional layer, followed by concatenation with feature maps from the corresponding down-sampling step. Thus, the model takes as input the image at time $t$ and the timestep $t$ and outputs the noise present in the image, as illustrated in Fig. 2. The model effectively captures spatial dependencies in the data and learns to predict the noise distribution based on both the current image and the timestep.

### III. RESULTS

The proposed model was initially utilized to generate 2D representations of ICNs. Subsequently, the model was adapted and extended to produce 3D ICNs. This progression from 2D to 3D generation allowed for a more comprehensive and detailed visualization of ICNs, capturing their spatial complexity and differences in their functional characteristics. The single-subject rs-fMRI data generates 53 ICNs through ICA each of them belonging to different functional domains. The dataset used for training the model was sourced from

the UK Biobank and comprised data from 10,000 subjects. Before training, we performed preprocessing steps such as normalization and background removal. Leveraging this large and diverse dataset, the model effectively captured variability across different networks, learning to generate realistic images that reflect the complex patterns of brain activity observed in each ICN.

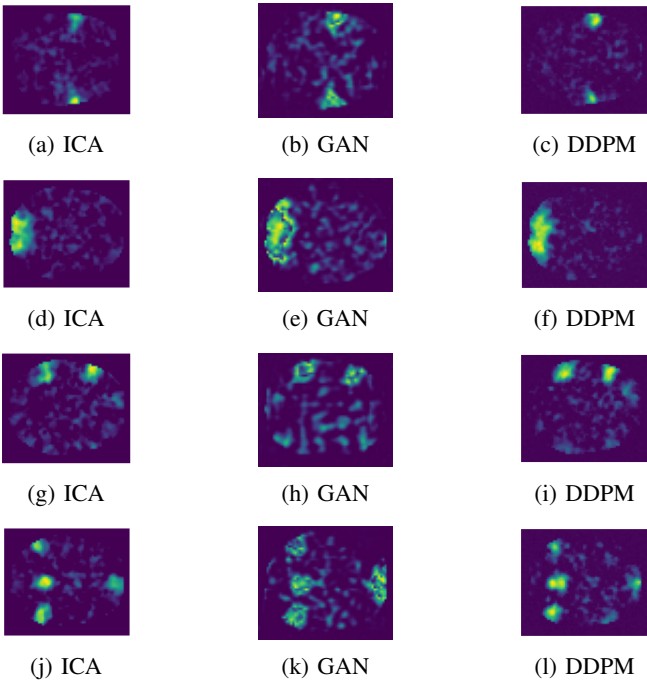

(a) ICA     (b) GAN     (c) DDPM

(d) ICA     (e) GAN     (f) DDPM

(g) ICA     (h) GAN     (i) DDPM

(j) ICA     (k) GAN     (l) DDPM

Fig. 3: Comparison of ICA, Progressive GANs and 2D-DDPM generated ICN images. The figure illustrates the accuracy and realism of the 2D-DDPM model in replicating the ICNs, highlighting the SM, VIS, CC, and DM domains.

The ICN images generated using DDPMs are compared with ICNs obtained from ICA, as illustrated in Fig. 3. While ICA is a linear method primarily designed for blind source separation, DDPMs are advanced nonlinear generative models. The comparison between them was done to highlight the specific strengths of DDPMs in capturing complex, nonlinear relationships within the data, which ICA may not fully address due to its linear nature. This also demonstrates the ability of DDPMs to generate more realistic and representative ICNs that better reflect underlying brain dynamics. Additionally, the model was compared with Progressive GANs [15], which, despite also being a nonlinear approach, resulted in inconsistent structural patterns in the generated ICNs. These likely occurred due to the challenges Progressive GANs face with mode collapse and instability during training. Unlike DDPMs, which employ a gradual noise addition and removal process that enhances stability and coverage of the data distribution, Progressive GANs struggle to accurately capture the complex variability inherent in ICNs. Furthermore, we extended our approach by training 3D-DDPMs on ICNs specifically from the Sensorimotor and Visual domains as shown in Fig. 4. These 3D models were

designed to capture the intricate three-dimensional spatial patterns of brain activity, providing a more detailed and comprehensive visualization of the connectivity networks. An assessment of the reconstruction quality of the 3D-DDPMs was performed through a numerical evaluation, utilizing metrics such as MS-SSIM (Multiscale Structural Similarity Index Measure), MSE (Mean Square Error), and PSNR (Peak Signal-to-Noise Ratio). They are standard metrics used in literature for assessing image quality, measuring structural similarity, and quantifying reconstruction error, respectively. MS-SSIM measures image similarity across various scales, encompassing luminance, contrast, and structure, with values ranging from -1 to 1. A score of 1 represents perfect similarity, while -1 indicates perfect dissimilarity. Hence it is crucial to ensure that the generated ICNs maintain the intricate structures of neuroimaging data. MSE and PSNR provide insights into the generated images' pixel-wise accuracy and signal fidelity. Together, these metrics offer a comprehensive evaluation of both the visual quality and quantitative accuracy of the generated ICNs.

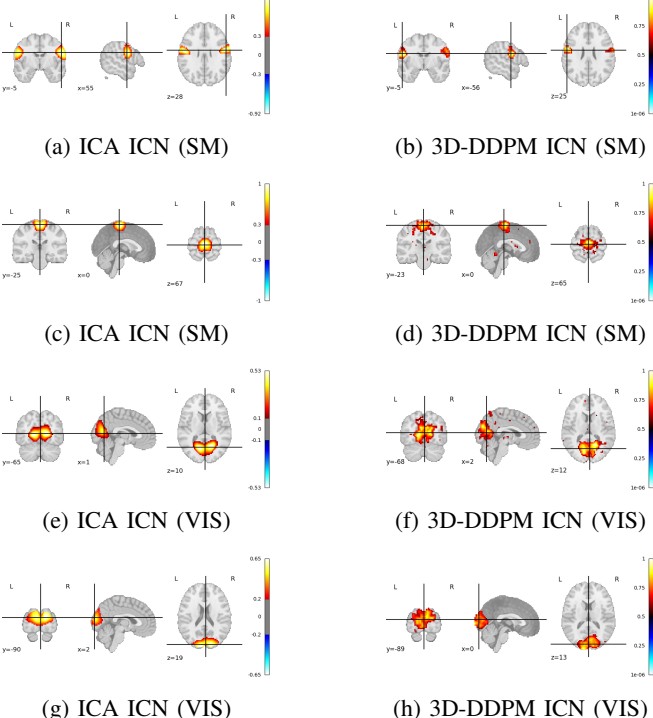

(a) ICA ICN (SM)     (b) 3D-DDPM ICN (SM)

(c) ICA ICN (SM)     (d) 3D-DDPM ICN (SM)

(e) ICA ICN (VIS)     (f) 3D-DDPM ICN (VIS)

(g) ICA ICN (VIS)     (h) 3D-DDPM ICN (VIS)

Fig. 4: Comparison between ICNs obtained from ICA and those generated by the 3D-DDPM model.

The study evaluates the model using both the UK Biobank and an independent dataset, specifically the Human Connectome Project (HCP) dataset [16], across all 53 ICNs. This approach helps mitigate the risk of overfitting associated with the complexity of DDPMs and ensures the generalizability of the findings to other datasets. After cross-validation with the UK Biobank dataset, the mean MS-SSIM was found to be 0.74, the mean MSE was 0.001, and the mean PSNR was 26.37. For the HCP dataset, the mean MS-SSIM was

0.72, the mean MSE remained 0.001, and the mean PSNR was 28.74. The results show that the model performs competitively on the HCP dataset, similar to its performance on the UK Biobank data, indicating that the model generalizes effectively.

Moreover, Fig. 5 depicts pixel-wise variance heatmaps derived from generated images. These heatmaps provide a visual representation of the variability in pixel values across 50 samples generated by the DDPM. High variance areas are indicated by cooler colors, exhibiting greater variability in pixel values across the generated images. Conversely, low variance areas are indicated by warmer colors, suggesting more consistent pixel values across the generated samples. In the heatmaps corresponding to the three different ICNs, the variance values range from 0.001 to 0.008. The narrow range of variance indicates relatively low variability in pixel values across the generated images, implying that the diffusion model consistently generates images with similar pixel values. Additionally, low variance in pixel values is also indicative of high-quality image generation. While this work currently uses only a few ICNs, it can be extended to all 53 ICNs to further demonstrate the model's versatility and robustness in generating accurate and realistic representations of brain connectivity networks.

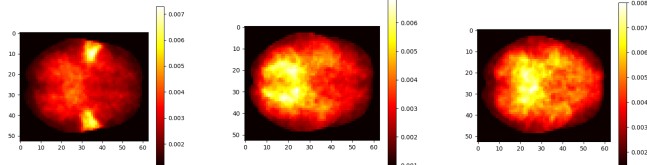

Fig. 5: Pixel-wise variance heatmap of three distinct generated ICNs. This heatmap highlights regions of high and low variability, with black/red indicating low variance and yellow/white indicating higher variability.

## IV. CONCLUSION

In conclusion, DDPMs offer a promising approach in the field of generative modeling in neuroimaging. By modeling the underlying probability distribution of data, DDPMs provide a powerful framework for generating high-quality samples. One of the key advantages of DDPMs is their versatility across dimensions, making them suitable for various types of data. This flexibility allows DDPMs to capture complex nonlinear dependencies in the data and generate realistic samples in different domains. The quantitative evaluation using various metrics also demonstrated the reconstruction capability of the DDPMs. However, scaling from 2D to 3D image generation using DDPMs presents significant computational challenges due to the increased data volume and complexity. Handling 3D data requires more memory, processing power, and sophisticated operations, leading to higher computational overhead. To address these challenges, strategies such as employing efficient 3D convolutional operations and utilizing distributed computing across multiple GPUs were implemented. These approaches collectively reduced the computational burden, making 3D generation more feasible and efficient. In future work, we will focus on subject-specific ICN generation using conditional DDPMs, allowing for a more tailored and individualized understanding of brain connectivity.

## ACKNOWLEDGMENT

This work was supported by the GSU RISE program and NSF grant 2316421.

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
