# OpenReview forum: "Denoising Diffusion Probabilistic Models for High-Fidelity fMRI Intrinsic Connectivity Network Data Generation"
_IEEE.org/EMBS/BHI/2024/Conference — IEEE BHI'24_

### Official Review · Reviewer_Xktm · 2024-08-13
**Denoising Diffusion Probabilistic Models for High-Fidelity fMRI Intrinsic Connectivity Network Data Generation**

**Overall Rating:** 7
**Confidence:** 4

**Other Quality Metrics:**

a. Fair
b. Good
c. Good
d. Fair

**Questions For The Authors:**

1. Can you provide more details on the challenges faced during the training of the DDPMs, particularly concerning the stability of the learning process?

2. Could you discuss any specific computational requirements or limitations encountered while scaling from 2D to 3D ICN generation?

3. How does the model handle the variability in ICNs across different subjects, and could this impact the generalizability of the synthesized ICNs?

**Strengths:**

1. The model's training on a large dataset tackles a common challenge in neuroimaging—sparse training data, enhancing its applicability in real-world scenarios.

2. The method includes a quantitative comparison of synthesized ICNs against ground truth, providing a good basis for assessing the model's performance.

**Summary Of The Paper:**

This paper introduces an rs-fMRI image synthesis framework leveraging denoising diffusion probabilistic models (DDPMs) to synthesize 3D intrinsic connectivity networks (ICNs).  The framework is trained on a large database to handle the sparsity of training datasets in neuroimaging, offering an advanced method for generating realistic connectivity patterns, which are evaluated against ground truth data.

**Weaknesses:**

1. While DDPMs provide advantages over traditional methods, they also introduce complexity in model training and interpretation, which could be a barrier in clinical applications.

2. The paper does not mention validation of the proposed method on external datasets, which is crucial for generalizability.

3. The paper lacks a detailed comparison with state-of-the-art methods, particularly other non-linear models that might offer competitive or superior performance.

---

### Official Review · Reviewer_Dg99 · 2024-08-15
**an innovative application of DDPMs but needs improvements in fair comparison, evaluation metrics clarification and rigorous validation**

**Overall Rating:** 7
**Confidence:** 4

**Other Quality Metrics:**

(a) Clarity of writing - good
(b) Clinical Significance - good
(c) Methodological Novelty - good
(d) Experiments and Results - good

**Questions For The Authors:**

1. I interpreted the comparison between DDPMs and ICA as possibly unfair because ICA is a linear method with a different primary purpose. It would be better to compare DDPMs with other advanced nonlinear methods or generative models specifically designed for similar tasks. However, if the authors intended to highlight specific strengths of DDPMs over ICA in this context, clarifying the rationale behind this comparison would be helpful.
2. It would be better to exchange the order of Figure 4 and 5.

**Strengths:**

1. The use of Denoising Diffusion Probabilistic Models (DDPMs) for generating high-fidelity intrinsic connectivity networks (ICNs) from rs-fMRI data is a novel approach.
2. The framework successfully generates both 2D and 3D ICN representations, providing detailed visualizations of brain connectivity. The ability to create 3D models is particularly valuable as it offers a more comprehensive view of brain activity.
3. The paper includes a thorough quantitative evaluation of the generated ICNs against ground truth data using metrics like MS-SSIM, MSE, and PSNR, adding credibility to the proposed method.

**Summary Of The Paper:**

The paper presents a framework for generating high-fidelity intrinsic connectivity network (ICN) data from resting-state functional MRI (rs-fMRI) using Denoising Diffusion Probabilistic Models (DDPMs). The authors propose that DDPMs, which capture complex nonlinear patterns, offer significant advantages over traditional linear methods like Independent Component Analysis (ICA). The framework progresses from generating 2D to 3D ICN representations, providing more detailed visualizations of brain connectivity. Trained on a large dataset from the UK Biobank, the model generates realistic ICNs, which are quantitatively evaluated against ground truth data, showing competitive performance in terms of accuracy and fidelity. The results suggest that DDPMs are effective for neuroimaging applications, especially in generating synthetic data to augment limited datasets and enhance research capabilities in brain connectivity studies.

**Weaknesses:**

1. The paper introduces DDPMs as a superior alternative to traditional methods like ICA for generating ICNs. However, the comparison might not be entirely fair. ICA is a well-established method for a specific purpose, and comparing it with DDPMs, which are fundamentally different and more complex, may not provide a balanced evaluation. It would be beneficial to include a comparison with other advanced nonlinear methods or generative models specifically designed for similar tasks. This could provide a more meaningful benchmark for evaluating the performance of the proposed framework.
2. While the paper uses quantitative metrics like MS-SSIM, MSE, and PSNR to evaluate the generated ICNs, the explanation of why these specific metrics were chosen and how they relate to the quality and fidelity of neuroimaging data is lacking. The authors could strengthen their argument by providing a clearer rationale for the selection of these metrics and discussing how they reflect the accuracy and utility of the generated ICNs in practical neuroimaging applications.
3. The study evaluates the model on a specific dataset from the UK Biobank and focuses on a limited number of ICNs. While the results are promising, the generalizability of the findings to other datasets and a broader range of ICNs is not fully addressed. The authors should consider validating their model on additional datasets from different sources and testing it across all 53 ICNs to ensure that the results are robust and applicable in various contexts.
4. Given the complexity of the DDPMs and the extensive training on a large dataset, there is a risk of overfitting, particularly if the generated ICNs closely resemble the training data. The authors should address how they mitigate this risk and provide evidence that the model can generalize well to unseen data. Techniques like cross-validation or testing on an independent dataset could help in demonstrating the model's generalization capabilities.

---

### Decision · Program_Chairs · 2024-09-23

Accept